# Self assembling cluster crystals from DNA based dendritic nanostructures

Emmanuel Stiakakis [1✉], Niklas Jung[2], Nataša Adžić [3], Taras Balandin [4], Emmanuel Kentzinger [5], Ulrich Rücker [5], Ralf Biehl [6], Jan K. G. Dhont[1,7], Ulrich Jonas[2] & Christos N. Likos [3✉]

Cluster crystals are periodic structures with lattice sites occupied by several, overlapping building blocks, featuring fluctuating site occupancy, whose expectation value depends on thermodynamic conditions. Their assembly from atomic or mesoscopic units is long-sought-after, but its experimental realization still remains elusive. Here, we show the existence of well-controlled soft matter cluster crystals. We fabricate dendritic-linear-dendritic triblock composed of a thermosensitive water-soluble polymer and nanometer-scale all-DNA den-drons of the first and second generation. Conclusive small-angle X-ray scattering (SAXS) evidence reveals that solutions of these triblock at sufficiently high concentrations undergo a reversible phase transition from a cluster fluid to a body-centered cubic (BCC) cluster crystal with density-independent lattice spacing, through alteration of temperature. Moreover, a rich concentration-temperature phase diagram demonstrates the emergence of various ordered nanostructures, including BCC cluster crystals, birefringent cluster crystals, as well as hex-agonal phases and cluster glass-like kinetically arrested states at high densities.

[1] Biomacromolecular Systems and Processes, Institute of Biological Information Processing (IBI-4), Forschungszentrum Jülich, D-52425 Jülich, Germany. [2] Macromolecular Chemistry, Department of Chemistry-Biology, University of Siegen, D-57076 Siegen, Germany. [3] Faculty of Physics, University of Vienna, Boltzmanngasse 5, A-1090 Vienna, Austria. [4] Structural Biochemistry, Institute of Biological Information Processing (IBI-7), Forschungszentrum Jülich, D-52425 Jülich, Germany. [5] Jülich Centre for Neutron Science JCNS and Peter Grünberg Institut PGI, JARA-FIT, Forschungszentrum Jülich, D-52425 Jülich, Germany. [6] Neutron Scattering and Biological Matter(JCNS-1/IBI-8), Forschungszentrum Jülich, D-52425 Jülich, Germany. [7] Heinrich-Heine-Universität Düsseldorf, Universitätsstraße 1, D-40225 Düsseldorf, Germany. ✉email: e.stiakakis@fz-juelich.de; christos.likos@univie.ac.at

Crystals are orderly states of matter in which particles with sizes ranging from sub-nanometer to micron are arranged in a periodic lattice. Crystalline solids epitomize the notion of rigidity, lying at the antipode of fluidity that is embodied by liquids. Accordingly, hybrid, exotic phases that combine crystallinity with (super-)fluidity have fascinated researchers both in the classical realm of soft matter physics[1–5] and in the quantum domain[6–10]. In usual crystals, the lattice constant $a$ and the particle concentration $c$ obey the proportionality $a \propto c^{-1/3}$, dictated by the condition that the (conventional) unit cell be populated by a fixed number of particles determined by the lattice geometry. Cluster crystals, a newer concept, are unconventional states of matter whose lattice sites are occupied by clusters of fully or partially overlapping particles rather than single ones[1–3,6–8,11,12]. In these states, the number of overlapping particles within a cluster, the lattice-site occupancy $N_{occ}$, is a fluctuating quantity, with its expectation value scaling with concentration as $N_{occ} \propto c$ and thus resulting in a concentration-independent lattice constant, the latter being the salient structural characteristic of both cluster crystals[1–3] and cluster quasicrystals[13,14].

Cluster crystals were first discovered in a simple model of penetrable spheres[11] and thereafter it was established that they are stabilized in general in any system of classical particles interacting by means of soft, bounded potentials whose Fourier transform has negative parts[1,3]. The interaction can be purely repulsive, leading thereby to the counterintuitive result of clustering in the absence of attractions[5,15]. This is a physical mechanism distinct to the one that leads to cluster formation in colloidal systems with diverging interactions combined with a strong, short-range attraction and a weak, long-range repulsion[16–19]. The clustering criterion has since then been generalized also to interactions featuring a hard core[20], and clustering phenomena have been experimentally observed in quasi-two-dimensional systems of core-softened magnetic colloids[21], whereas similar phenomenology has been observed in simulations of cell colonies[22]. Concrete suggestions for realizing cluster-forming building blocks in the soft matter have been made in computer simulation models on the basis of effective potentials[23], and for concentrated solutions in full, monomer-resolved simulations[6,24,25].

Soft matter cluster crystals bear striking analogies with the supersolid state of matter in the quantum regime[8–10]. Scientific breakthroughs in atomic physics have made it possible to create artificial interatomic potentials by exploiting collective matter-light interactions in cavities[26,27] or by weakly coupling a Rydberg state to the ground state using laser light[28–32]. Theoretical investigations have demonstrated that the resulting core-softened, repulsive interactions stabilize cluster- and supersolid phases with remarkable similarities to soft matter cluster crystals[29,30,33–38]. There has been growing experimental evidence for the existence of (metastable) quantum cluster- or supersolids in the last few years[39–41] but no experimental realization of the soft matter cluster crystals has been reported thus far. Here, we show theory-informed, suitably designed DNA-based dendritic triblock are appropriate soft-matter building blocks unambiguously leading to the formation of stable cluster crystals whose properties conform to earlier theoretical predictions.

## Results and discussion

**Building blocks design.** We synthesized and studied the self-assembly of DNA-based dendritic-linear-dendritic triblock. Neutral bifunctional Poly(2-oxazoline)-based copolymers (Poxa) chains that possess lower critical solution temperature behavior ($T_{LCST} \cong 33\,°C$ in 150 mM NaCl aqueous salt solutions) were end-capped with all-DNA charged stiff dendrons[42,43] of first and second generation (Fig. 1). Poxa is a thermoresponsive polymer that exhibits a reversible and sharp coil-to-globule phase transition in water by increasing the temperature above the $T_{LCST}$[44]. The conformational change of the individual Poxa chains is accompanied by partial dehydration, suggesting that Poxa alters hydrophilicity and hydrophobicity abruptly in the vicinity of $T_{LCST}$ (see measurements of the cloud point of Poxa in Supplementary Method 3). The dendrons' free-ends were terminated by a non-sticky single-stranded DNA (ssDNA) dangling tail in order to ensure that possible intermolecular base-stacking interaction between blunt-ended DNA helices is prohibited[45]. We refer to the DNA-based triblock build up from first and second-generation dendrons as the G1-P-G1 (Fig. 1a) and G2-P-G2 (Fig. 1b), respectively. Non-denaturing gel electrophoresis was employed to confirm the successful assembly of the DNA-polymer architectures which are schematically depicted in Fig. 1a–c (see Fig. 1d). The gyration radii $R_g$ are 6.4 nm and 9.8 nm for G1-P-G1 and G2-P-G2, respectively. More details regarding the synthesis and the molecular characterization are given in the Methods, Supplementary Methods 1–3, and Supplementary Note 1.

Our choice of the experimental building blocks is guided by computer-based design ideas for dendritic-type molecules whose effective interactions satisfy the prerequisites for cluster crystal formation[6,23,25]. The (electrostatically) repulsive dendrimer coronae give rise to a repulsive interaction, which increases as the separation between the dendrimers' centers of mass diminishes. This trend is tempered by the effective mutual attraction between the thermosensitive polymers in the core of the molecule. In this way, a core-softened repulsion results[6,23,25]. Due to the decreasing solubility of bare Poxa at higher temperatures, we expect that the propensity to cluster formation will become stronger as the temperature increases[23]. Similarly, since the electrostatic repulsion between the terminal DNA-dendrons is enhanced at higher generations, the clustering ability of the G2-P-G2 blocks is anticipated to be weaker than that of their G1-P-G1 counterparts. The key structural features of the proposed dendritic-like building blocks are their open structure and the chemical dissimilarity between the flexible Poxa chains

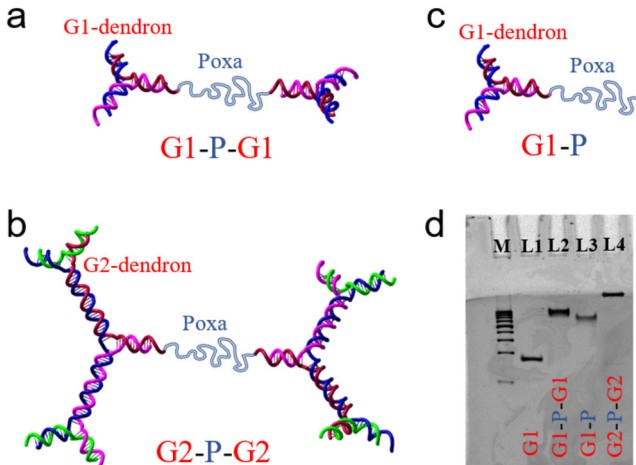

**Fig. 1 Schematics and characterization of the tethered all-DNA dendrons. a** First generation of dendritic-linear-dendritic triblock (G1-P-G1). **b** Second generation of dendritic-linear-dendritic triblock (G2-P-G2). **c** First generation of dendritic-linear diblock (G1-P). **d** Non-denaturing polyacrylamide gel electrophoresis (PAGE) analysis. 10% PAGE: Lane M contains 50 base-pair (bp) DNA markers. Lanes 1–4 contain G1, G1-P-G1, G1-P, and G2-P-G2, respectively.

and the stiff all-DNA dendrons. By linking together two stiff and highly charged dendritic blocks (all-DNA dendrons) through a long and flexible polymer chain (Poxa), in combination with the microphase separation mechanism driven by the immiscibility between the DNA and polymeric blocks[46], we allow the centers of masses of different DNA-based constructs to coincide, without significant interpenetration of the dendrons. We shall demonstrate that the degree of segregation between all-DNA dendron and Poxa blocks can be temperature-controlled within the weak limit[47], owning to Poxa's LCST and triblock's macromolecular architecture. The beneficial effect of this is reflected by the emergence of stimuli-responsive cluster crystals by altering the temperature, leading to intriguing phase transition pathways. All experiments were carried out in 1xTris/Na buffer (10 mM Tris, pH: 8.0, 150 mM NaCl).

**Absence of a micellization mechanism in DNA dendritic-based triblock.** The tendency of particles to form stable clusters in the absence of attractions is a phenomenon clearly distinct from micellization, which is common for block copolymers in selective solvents[48–50], polymeric amphiphiles[51–57], and small molecule amphiphile surfactant sytems[58]. A conventional block-copolymer amphiphilie, a system most relevant to our DNA dendritic-based triblock, is commonly composed of a hydrophilic and a hydrophobic segment that are covalently linked. The hydrophobic part can be a synthetic polymeric block[51,52,54,59] or different types of moieties (such as long-carbon alkyl chains, lipid molecule and fluorescent dyes)[53,55,57]. These amphiphiles can be assembled into micelles (aggregates) with rich morphological and size diversity at room temperature at low critical micelle concentration (cmc); with the latter obtained at extremely low concentrations, and in particular many orders of magnitude below the overlap concentration ($c^*$) of amphiphiles in solution. In addition, temperature-dependent hydrophobic blocks, similar to the Poxa employed in this study, can result in a thermoresponsive cmc, allowing micelle assembly and disassembly upon a change in temperature. This means that the segregation strength between the blocks forming this type of polymeric amphiphiles can be externally controlled, allowing access from the weak- to strong-segregation regime[47].

However, the segregation strength of the above-mentioned system and consequent its cmc behavior is strongly dependent on the position of the thermoresponsive block relative to the hydrophilic block in the block-copolymer amphiphilie. By comparing the dilute self-assembly behavior of the dendritic-based triblock (G1-P-G1) and diblock (G1-P), we show that the encapsulation of the Poxa block in an effective shell of two all-DNA dendrons results in the absence of micellar aggregates at temperatures well above the $T_{LCST}$ of Poxa. Static light scattering (SLS) and dynamic light scattering (DLS) measurements were employed to determine the presence and hydrodynamic radius ($R_H$) of aggregates. Figure 2 presents the temperature-dependent self-assembly behavior of the G1-P-G1 and its linear-dendritic analog (G1-P, Fig. 1c) in dilute aqueous solutions at a NaCl concentration of 150 mM containing buffer (Methods and Supplementary Note 1). For the G1-P system (red-symbols in Fig. 2), the temperature dependence of $R_H$ and the SLS intensity $I$ (at fixed scattering angle $\theta = 90^o$, $q = 0.0187$ nm$^{-1}$) indicate the formation of large aggregates with a narrow distribution in size at a temperature slightly above the Poxa's $T_{LCST}$ ($R_H = 175.3$ nm, see Supplementary Fig. 1a). Such a molecular aggregation is clearly absent in the case of G1-P-G1, as illustrated in Fig. 2 (black-symbols); with the scattering intensity and hydrodynamic radius to be virtually unaffected within the range of 15 °C to 50 °C (see also DLS data for the G2-P-G2 at temperature well-above the

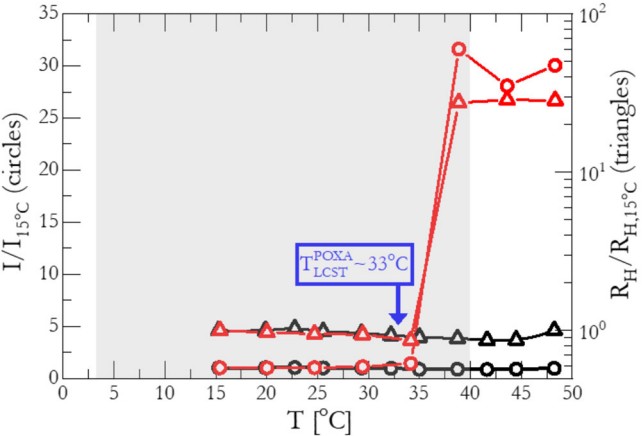

**Fig. 2 Dilute solution properties of G1-P-G1 and G1-P.** Temperature-dependent static light scattering (SLS) intensity (left-axis, circles) at a scattering angle of $\theta = 90$ °, and hydrodynamic radius ($R_H$, right-axis, triangles), normalized with respect to their values at $T = 15$ °C, of dilute G1-P-G1 (black curves) and G1-P (red curves) aqueous solutions (with DNA concentration $c$ equal to 5.0 mg/ml and 3.0 mg/ml, respectively) using 1xTris/Na buffer (10 mM Tris-HCl, pH: 8.0, 150 mM NaCl). The gray-zone indicates the temperature window where the concentrated G1-P-G1 and G2-P-G2 self-assembled phase behavior is investigated.

$T_{LCST}$ in Supplementary Fig. 1a). In full agreement with the LS data, the G1-P-G1 and G2-P-G2 form-factors, as probed by small-angle X-ray scattering (SAXS) experiments reveal that their global size (radius of gyration, $R_g$) and internal structure is temperature-insensitive (see Supplementary Fig. 1b, c). The $R_g$ and $R_H$ values of the G1-P-G1 and G2-P-G2 systems are listed in the Methods, ("System parameters" section).

From the above results, it becomes evident that the temperature-dependent solvophobicity of the Poxa-block does not act as an effective short-range attraction in the interaction potential of our DNA-based triblock that could initiate their aggregation into stable clusters. The lack of a critical concentration and/or temperature for the G1-P-G1 and G2-P-G2 micellization is intimately related to the absence of enthalpically driven aggregation processes. However, the change of temperature allows the external steering of the microphase separation between the two chemically incompatible blocks forming the proposed DNA dendritic-based triblock, but only within the weak-segregation limit. We shall demonstrate that this property will emerge as a key factor for the rich phase behavior of G1-P-G1 and G2-P-G2 which is encountered at DNA concentrations above their overlap values $c^*$ (see methods, "system parameters" section).

In the present case, the building blocks have a repulsive effective interaction with one another and the formation of clusters occurs at concentrations close to their overlap value, in agreement with theoretical predictions[11]. The normal liquid thus crosses over gradually to a cluster liquid[15,24] as the overlap density is approached, and at even higher concentrations the cluster fluid undergoes a first-order phase transition to the cluster crystal[2,3]. This self-assembly pathway is sketched in Fig. 3a–c by depicting representative simulation snapshots from the model of Ref.[2]. Together, in Fig. 3d–f, the corresponding concentration-dependent cluster-forming scenario for the G1-P-G1 system is depicted.

**Phase diagrams of cluster-forming dendrimers.** SAXS measurements served as the basis for determining the phase behavior of the dendritic-like constructs presented in Fig. 1a–b. A

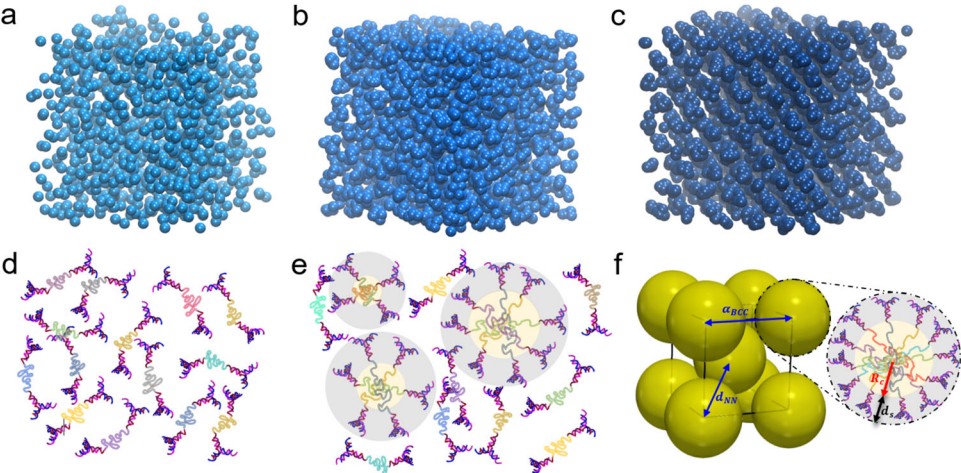

**Fig. 3 Simulation snapshots and schematic representations of cluster-forming particles at different concentrations.** Top panel: the snapshots were produced by performing Monte Carlo simulations of point particles interacting by the model, cluster-forming potential $v(r) = \varepsilon \exp[-(r/\sigma)^4]$ at temperature $k_B T/\varepsilon = 0.6$, and densities $\rho\sigma^3$ as follows: **a** $\rho\sigma^3 = 1.0$, where a normal fluid is stable; **b** $\rho\sigma^3 = 3.8$, where a cluster fluid forms; **c** $\rho\sigma^3 = 4.1$, resulting into a cluster body-centered cubic (BCC) crystal. Each sphere represents a single building block. Bottom panel: schematic representation of the proposed concentration-dependent mechanism for the G1-P-G1 clustering in the absence of attractions, and, finally, the G1-P-G1 clusters crystallization into a BCC cluster crystal structure: **d** G1-P-G1 fluid; **e** polydisperse G1-P-G1 clusters with a core-shell architecture, in coexistence with non-aggregated species; **f** BCC crystal formed by monodisperse G1-P-G1 clusters. The characteristic lengths are also illustrated ($R_c$: cluster core radius; $d_s$: cluster shell thickness; $a_{BCC}$: BCC lattice constant; $d_{NN}$: the nearest neighbor distance along with the room diagonal). Different colors for the Poxa block are added to assist in identifying individual G1-P-G1 molecules.

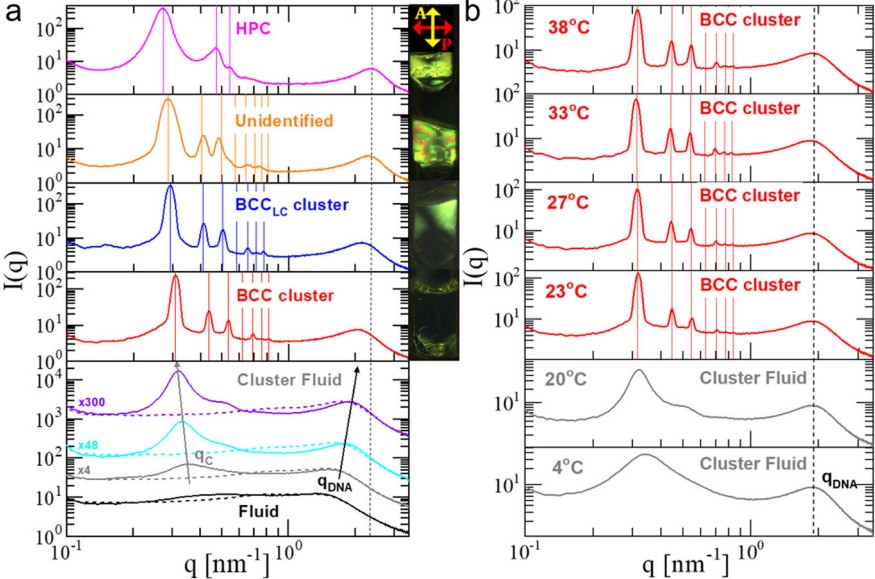

**Fig. 4 Self-assembly of G1-P-G1. a** 1D-SAXS profiles at 20 °C for the G1-P-G1, with DNA concentrations from the bottom to the top panel (solid-lines): 156.5 (black), 188.2 (gray), 216.9 (cyan), 255.7 (violet), 287.7 (red), 322.9 (blue), 337.7 (orange) and 356.2 (magenta) mg/ml. The corresponding image of the sample between cross-polarizers is also illustrated for selected concentrations at the right side of the panels. In the bottom panel, the 1D-SAXS profiles for the G1 system (dashed-lines) are included, at the same total DNA concentration as for the G1-P-G1. Also, selected profiles are shifted along the intensity axis for clarity. The intensity shift factors are presented on the left side of the profiles. In the second panel from the top, the expected reflections for a BCC lattice based on the position of the first intense peak in the 1D-SAXS profile are shown. **b** Temperature-dependent SAXS profiles of the G1-P-G1 at concentration $c = 255.7$ mg/ml. For each 1D-SAXS profile, the corresponding type of phase behavior is indicated (BCC: Body-Centered Cubic with $Im\bar{3}m$ space group symmetry, $BCC_{LC}$ cluster: Liquid Crystalline BCC-like cluster crystal, HPC ($P6/mm$): Hexagonally Packed Cylinders with $P6/mm$ space group symmetry. The vertical solid-lines indicate the positions of the first seven possible reflections for the stated morphologies. The most right black dashed-line is guide for the temperature- and concentration-dependence of the $q_{DNA}$-peak.

representative series of one-dimensional (1D)-SAXS patterns at 20 °C, a temperature well below the Poxa's LCST, is shown in Fig. 4a for various concentrations of aqueous solutions of G1-P-G1 containing 150 mM NaCl. By increasing the DNA concentration, a rich phase transition pathway from disordered to

various ordered phases is observed. Over the concentration range 156.5 mg/ml ≤ c ≤ 255.7 mg/ml (bottom panel of Fig. 4a), the SAXS patterns indicate that the G1-P-G1 solution undergoes gradually a transition from a usual fluid to a disordered state of clusters with strong positional correlations between the dendritic-

based building blocks. The latter is evidenced by the presence of a pronounced peak with the maximum of the scattered intensity located around the scattering wave vector value $q_c \cong 0.33$ nm$^{-1}$, which corresponds to a spatial correlation value $d_c = 2\pi/q_c \cong 19.0$ nm and represents the average inter-cluster separation. The prediction from simulation[24,25] is $d_c \cong 3R_g$; using the value $R_g = 6.4$ nm for the G1-P-G1 blocks at hand (Methods and Supplementary Note 1), we obtain $d_c = 19.2$ nm, in excellent agreement with the experiment. The SAXS profile in the fluid phase (Fig. 4a, bottom panel, black solid-line, $c = 156.5$ mg/ml) displays a weak and broad Bragg reflection at $q > 1.0$ nm$^{-1}$. The position of this correlation peak, $q_{DNA}$, is related to the average distance between neighboring DNA helices[60,61]. Above the fluid phase, at DNA concentration $c = 188.2$ mg/ml (Fig. 4a, bottom panel, gray solid-line), the SAXS pattern shows the existence of an additional broad peak $q_c$, which becomes progressively more intense and narrow with increasing G1-P-G1 concentration (Fig. 4a, bottom panel, cyan and purple solid-lines); simultaneously the peak moves to noticeable lower $q$-values.

These observations evidence the presence of stable clusters consisting of finite aggregates of G1-P-G1 molecules, as their concentration approaches its overlap value, $c^* \cong 196.0$ mg/ml (see Methods, "System parameters" section). The existence of such a cluster fluid phase is further corroborated by the SAXS patterns obtained from the G1 system, at the same total DNA concentrations as for the G1-P-G1 system (Fig. 4a bottom panel, dashed-line profiles), which reveal the absence of such cluster-induced interference peak. This allows us to assign the $q_c$ peak to cluster-cluster correlations mediated by electrostatic interactions between the charged clusters, whereas the $q_{DNA}$-peak, contrary to the fluid phase, corresponds to positional correlations of the DNA segments within a single cluster. Such a peak interpretation can offer a plausible explanation for the opposite concentration dependence of the $q_{DNA}$- and $q_c$-peak shifts within the cluster fluid regime (bottom panel, black and gray arrows, Fig. 4a). The very open structure of our dendritic-based macromolecules allows the centers of masses of different G1-P-G1 to lie on top of each other, thus forming spherical aggregates with a core-shell architecture, with the DNA dendrons (G1) located within the shell region (see Fig. 3e). On increasing the DNA concentration, clusters with such an internal structure can progressively grow by further aggregation, however, with a slight change in their overall size avoiding significant overlapping of the charged dendrons. The above intra-cluster packing scenario implies a simultaneous increase of the cluster's shell DNA density and charge. Both effects are reflected in the $q_c$- and $q_{DNA}$-peak shifts with increasing concentration (see arrows at the bottom panel of Fig. 4a), respectively. The stability of the cluster fluid phase is in agreement with theoretical predictions[2,15,24].

At the vicinity of the G1-P-G1's overlap density, the above clustering-forming mechanism is spontaneously triggered to counter high levels of packing frustration owing to excluded-volume constraints at the molecular level[6,25]. A complete overlap and penetration of a dendritic-linear-dendritic triblock with a few others becomes energetically preferable over partial overlaps with many neighbors as a consequence of its open-internal structure. Additionally, the entropic penalty that the Poxa chains experience as they possibly overstretch in an attempt to increase the aggregation number inside the cluster is effectively counter-balanced by the inherent tendency of the flexible Poxa and the stiff all-DNA dendrons to minimize their unfavorable contacts at the inter-cluster level, similar to classical block copolymers[62].

The cluster fluid phase persists up to a DNA concentration of 255.7 mg/ml (Fig. 4a, violet solid-line), however, with the $q_c$-peak accompanied by an emerging, broad peak at slightly higher $q$-values. This indicates that the cluster fluid phase becomes more

structured, signaling the onset of a disorder-to-order transition by increasing the G1-P-G1 density. Indeed, strikingly different is the appearance of small-angle ($q < 1.0$ nm$^{-1}$) scattering peaks for a DNA concentration of 287.7 mg/ml, as can be seen from the second panel from the bottom in Fig. 4a. An intense and sharp principal peak, at a scattering wave vector value $q^* \cong 0.31$ nm$^{-1}$, slightly below the $q_c$-peak, and several higher-order reflections appear, offering an unambiguous determination of the lattice structure and lattice constant based solely on the location of these Bragg peaks. The finding $q^* \lesssim q_c$ is consistent with the predictions from simulations[2,6,25] that upon ordering on a crystalline arrangement, the clusters become less polydisperse and more compact, thereby slightly increasing their inter-cluster separation. The corresponding SAXS profile displays six distinguishable peaks centered at $q/q^* = 1 : \sqrt{2} : \sqrt{3} : \sqrt{5} : \sqrt{6} : \sqrt{7}$ (vertical red lines), providing conclusive evidence that the G1-P-G1 clusters have self-assembled into a body-centered cubic (BCC) crystal (space-group $Im\bar{3}m$) with significant long-range order and a conventional unit cell of size $a_{BCC} = \sqrt{2}(2\pi/q^*) = 28.7$ nm (see lattice topology in the right cartoon of the bottom panel in Fig. 3c. The faint presence of the fourth reflection at $q/q^* = \sqrt{4}$ is due to the minimum in the cluster's form-factor (blue curve in Supplementary Fig. 2a and discussion in Supplementary Notes 2.2–2.3). The reported cubic phase is also attested by the absence of optical birefringence, as clearly illustrated in the corresponding depolarized image of Fig. 4a (second panel from the bottom, right image). Upon further compression ($c = 322.9$ mg/ml, third panel from the bottom in Fig. 4a), the 1D-SAXS profile shows that all the reflections are preserved and shifted to lower $q$-values. The blue vertical lines above the intensity profile indicate the locations of Bragg peaks with a BCC lattice constant of 30.7 nm. However, surprisingly, the corresponding depolarized image of the sample reveals a counter-intuitive uniform birefringence (Fig. 4a liquid crystalline BCC-like, BCC$_{LC}$), since one would expect that a cubic phase will be optically isotropic.

Ultimately, increasing the G1-P-G1 concentration leads to a number of significant changes in the scattering and birefringence patterns. At DNA concentration $c = 337.7$ mg/ml, the sample under crossed polarizers shows a vivid colorful birefringence, with the principal $q^*$-peak becoming noticeably wider; it also shifts to lower $q$-values (second panel from the top, Fig. 4a). In contrast, the shape of the higher order reflections is distorted, with their positions being concentration-independent. The above observations, in combination with the clear disagreement between the observed peaks and the allowed reflections (orange vertical lines) for a BCC crystal based on the primary and most intense reflection at $q^*$, are indicative for a mechanically unstable crystal phase. Eventually, upon further increase of the DNA concentration to $c = 356.2$ mg/ml, (top panel of Fig. 4a), the G1-P-G1 solution undergoes a structural phase transition to a hexagonal packed cylinders morphology (HPC with P6/mm space-group symmetry), as evidenced by the three clear reflections at ratios $q/q^* = 1 : \sqrt{3} : \sqrt{4}$ (magenta vertical lines). From the primary peak, the inter-column distance $d_{HPC} = 4\pi/(\sqrt{3}q^*)$ is calculated to be 26.8 nm.

The spontaneous cluster crystallization is clearly demonstrated by the temperature-dependent 1D-SAXS profiles of the G1-P-G1 at DNA concentration $c = 255.7$ mg/ml, which are presented in Fig. 4b. By increasing the temperature, a phase transition from (cluster) fluid to BCC crystal with $a_{BCC} = 28.2$ nm is observed, similar to the concentration-dependent manner presented in the two bottom panels of Fig. 4a (a more detailed discussion regarding the SAXS profiles of Fig. 4b is presented in Supplementary Note 3). It is particularly important to emphasize

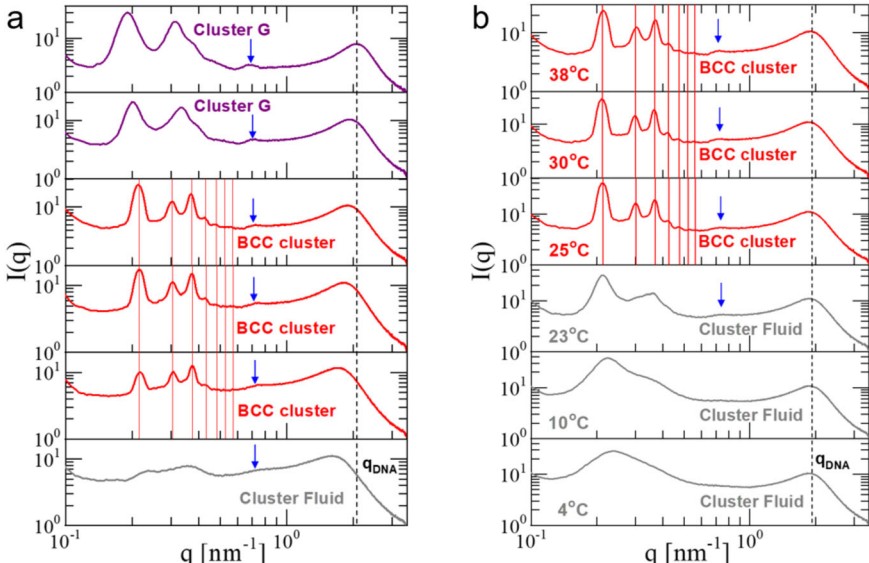

**Fig. 5 Self-assembly of G2-P-G2. a** 1D-SAXS profiles at 38 °C for the G2-P-G2, with DNA concentrations from the bottom to the top panel: 177.2 mg/ml (gray); 190.8, 204.7, 220.5 mg/ml (red); 234.08, 250.6 mg/ml (purple). **b** Temperature-dependent SAXS profiles of the G2-P-G2 at a concentration $c = 220.5$ mg/ml. The blue arrows denote the presence of the cluster form-factor feature in the corresponding 1D-SAXS profiles (see also Supplementary Fig. 2). For each 1D-SAXS profile, the corresponding type of phase behavior is indicated (BCC: body-centered cubic with $Im\bar{3}m$ space group symmetry and Cluster G: Non-birefringent Glass-like. The vertical solid-lines indicate the positions of the first seven possible reflections for the stated morphologies. The most right black dashed-line is guide for the temperature- and concentration-dependence of the $q_{DNA}$-peak.

that this fluid-to-crystal transition occurs within a narrow temperature range (between 20 and 23 °C) which is noticeably lower than the LCST of Poxa. Interestingly, heating the sample up to 38 °C, a temperature well above the LCST of Poxa, produces identical scattering patterns compare to those at 23 °C. The latter is in accordance with the temperature-independent behavior of G1-P-G1 in dilute conditions. Therefore, the results in Fig. 4b undoubtedly supports our claim for the absence of a micellization mechanism in the reported DNA-based dendritic building blocks.

The influence of dendron generation on the clustering ability of these DNA-based dendritic-linear-dendritic triblock and their subsequent self-assembly to cluster crystals is demonstrated in Fig. 5, where the concentration- and temperature-dependent phase behavior of G2-P-G2 (Fig. 1b) are presented. Both encompass the same type of cluster fluid-to-crystal transition, with the increase of dendron size to be consistently reflected in the measured BCC lattice constant, $a_{BCC} = 42.0$ nm at DNA concentration $c = 220.5$ mg/ml (fourth panel from the bottom of Fig. 5a and, at for temperatures above 23 °C, Fig. 5b). Also, the BCC crystal shows temperature- and concentration-independent lattice constants, as witnessed by the Bragg peaks positions over a relatively broad range of DNA densities (190.8 ≤ $c$ ≤ 220.5 mg/ml at $T = 38$ °C, Fig. 5a) and temperatures (25 °C ≤ $T$ ≤ 38 °C at $c = 220.5$ mg/ml, Fig. 5b). However, contrary to the G1-P-G1 where the concentration increase revealed a rich phase transition pathway from cluster fluid to various crystalline phases, the G2-P-G2 BCC cluster crystal becomes unstable at high DNA densities, leading to a disordered solid. This is illustrated in the scattering profiles of the two top panels of Fig. 5a, both of which demonstrate that the sharp principal peak and higher orders reflections are replaced by two broad peaks which systematically move toward lower $q$-values with increasing concentration. These intensity profiles demonstrate an intriguing similarity with the one obtained from the cluster fluid phase (Fig. 5b, $T = 23$ °C), suggesting that the observed disordered phases are reminiscent of cluster glass-like structures (Cluster G). These findings are in agreement with simulation results based on monomer-resolved

models[6,25]. Indeed, in contrast to simplified models based on pairwise additive effective interactions[3], for real dendrimers the cluster population cannot grow indefinitely with concentration and thus the clean theoretical prediction $N_{occ} \propto c$ eventually breaks down at some system-specific crossover concentration $c_\times$. For values $c \gtrsim c_\times$, distortions of the lattice constants, anisotropic phases and even glassy behavior have also been seen in simulations of such systems[6,25]. Interestingly, the cluster-forming SAXS signature can be easily detected in the 1D-SAXS profiles of Fig. 5a as indicated by the blue arrows, suggesting that clusters persist over the concentration-dependent phase transition pathway from cluster fluid to cluster BCC crystal and finally to a disordered solid. Finally, it is important to underline that the disorder-to-order cluster transition is fully reversible with temperature (Supplementary Fig. 3), and therefore it reflects thermodynamic equilibrium morphologies.

Further analysis of the 1D-SAXS profiles corresponding to the G1-P-G1 (Supplementary Fig. 2a) and G2-P-G2 (Supplementary Fig. 2b) BCC cluster crystals show that the global cluster size of the G1-P-G1 and G2-P-G2 is noticeably larger than the radius of gyration of a single particle, implying the formation of clusters with moderate occupancy (details on the fitting procedure of 1-D SAXS profiles are given in Supplementary Notes 2.1–2.2). A schematic of the cluster model is depicted in Fig. 3c (bottom panel, right cartoon). Indeed, from the total DNA concentration and BCC lattice constant, absolute values of an average cluster occupancy number $N_{occ}$ can be estimated (see Methods, "System parameters" section), leading to $N_{occ} = 29$ for both G1-P-G1 and G2-P-G2. In addition, the fitting values for the nearest neighbor distance $d_{NN}$ along the body diagonal of the formed BCC crystals (left cartoon in Fig. 4f) indicate that the dendritic-based cluster crystals almost reach the close-packed configuration (see table for the fitting parameters in Supplementary Note 2.1 and relevant analysis in the Supplementary Note 2.2). The lattice constants correspond to a ratio $a_{BCC}/R_g \cong 4.3$ (27.9 nm/6.4 nm and 42.0 nm/9.8 nm for G1-P-G1 and the G2-P-G2, respectively), in excellent agreement with simulations of generic similar models[25],

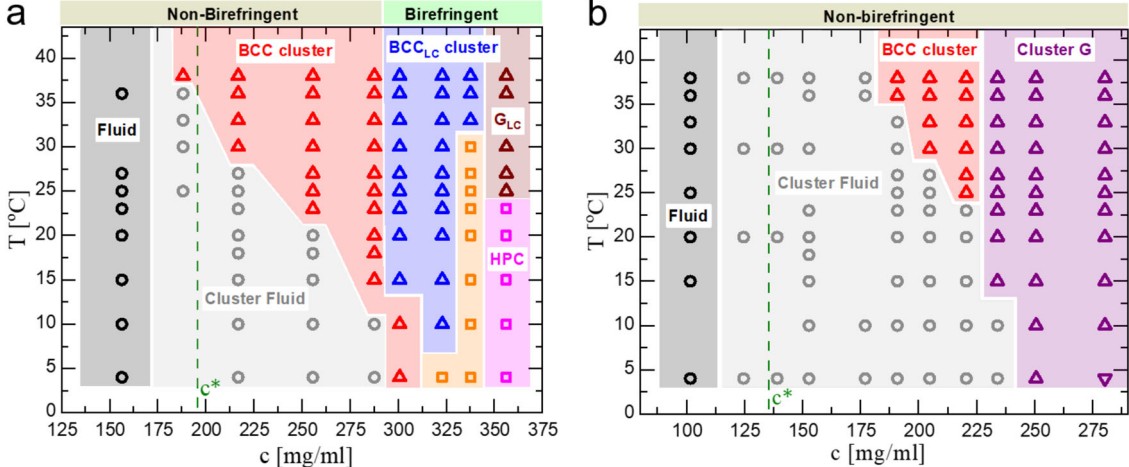

**Fig. 6 Phase diagrams of G1-P-G1 and G2-P-G2.** A concentration-temperature phase diagram of aqueous solutions of G1-P-G1 (**a**) and G2-P-G2 (**b**). The following phases are indicated: Fluid (black circles), cluster fluid (gray circles), BCC cluster crystal (red triangles), liquid crystalline BCC-like cluster crystal ($BCC_{LC}$ cluster, blue triangles), liquid crystalline glass-like ($G_{LC}$, brown triangles), non-birefringent glass-like (Cluster G, purple triangles) and hexagonal packed cylinder (HPC, magenta squares). The corresponding background colors are added to assist in identifying the various phases. The structural assignment of the orange region in the G1-P-G1 phase diagram based solely on the SAXS data was not possible. The green-dashed lines indicate the DNA overlap concentration $c^*$ of G1-P-G1 and G2-P-G2 (see Methods, "System parameters" section).

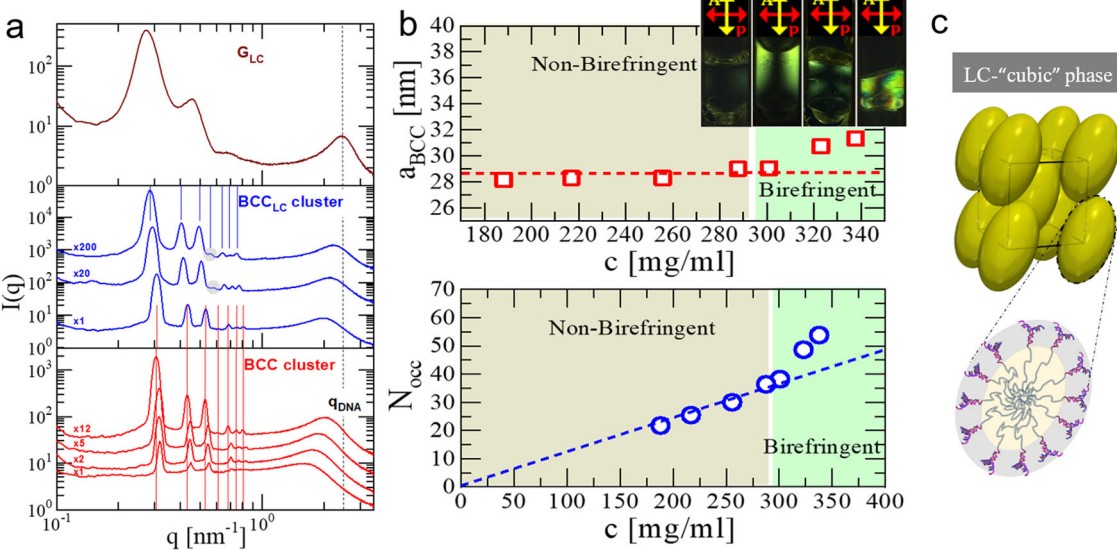

**Fig. 7 Cluster phases and extracted lattice parameters at various concentrations. a** 1D-SAXS profiles at 38 °C for the G1-P-G1, with DNA concentrations from the bottom to the top panel: 188.2, 216.9, 255.7, 287.7 (red profiles), 300.8, 322.9, 337.7 (blue profiles) and 356.2 (brown profile) mg/ml. The vertical solid-lines indicate the positions of the first seven allowed reflections for a BCC crystal. In the middle panel, the gray circles indicate the appearance of the fourth diffraction peak. The black dashed-line is guide for the concentration-dependence of the $q_{DNA}$ peak. The profiles presented in the bottom and middle panels are shifted along the intensity axis for clarity. The intensity shift factors are presented on the left side of the profiles. **b** The BCC lattice constant $a_{BCC}$ (top panel) and the cluster occupancy $N_{occ}$ (bottom panel) against G1-P-G1 concentration at 38 °C. The dashed lines are guides for the concentration-dependence of $a_{BCC}$ (red-line) and $N_{occ}$ (blue-line) within the non-birefringent regime. Top panel inset: Depolarized images of selected G1-P-G1 samples at 38 °C. DNA concentration from the left to the right: 287.7, 300.8, 322.9 and 337.7 mg/ml. **c** Schematic of the proposed arrangement and deformation of the G1-P-G1 clusters within the optically anisotropic BCC-like phase (Liquid Crystalline (LC)-"cubic" phase).

in which the ratio $a_{BCC}/R_g \cong 4.1$ was found for broad density ranges.

The phase diagrams of G1-P-G1 and G2-P-G2 as a function of the temperature and the total DNA concentration are presented in Fig. 6. A combination of the above-mentioned SAXS measurements and depolarized images of the samples allow us to identify most of the formed phases. Both systems, at sufficiently high densities, form stable clusters in the fluid phase, which upon further compression self-organize into cluster BCC

crystals in a temperature-dependent manner. The independence of the peak positions on concentration, the bottom panel of Fig. 7a, points to the density-independent lattice constant, top panel of Fig. 7b. In contrast to the G1-P-G1 case, the phase diagram of G2-P-G2 displays one type of crystal phase (BCC cluster) with its thermodynamic stability restricted in a noticeable narrower region in the plane of temperature versus concentration (red areas in Fig. 6a-b), as previously anticipated. While G2-P-G2 exhibits only an optically isotropic glass-like (Cluster G, purple

area in Fig. 6b and SAXS profiles in Fig. 5a) behavior at higher DNA densities, the G1-P-G1 demonstrates a rich structural polymorphism involving birefringent crystals and birefringent glass-like phases (G$_{LC}$, brown region in Fig. 6a, and SAXS profile in the top panel of Fig. 7a). This is not surprising given the increased dendron size which is employed for the fabrication of the G2-P-G2 triblock. Although the underlying physical mechanism that drives the emergence of clustering is identical for G1-P-G1 and G2-P-G2, the latter encounters higher levels of dendron packing frustration since the length of the polymeric tether (Poxa chain) has remained unchanged. The correct length of the Poxa chain can lead to not only a widening of the region where the BCC cluster crystal is stable but also allow liquid crystalline morphologies (BCC$_{LC}$ and HPC in Fig. 6a) to become stable. Therefore, the G2-P-G2 phase diagram highlights the key role of the open architecture of the building blocks for the stabilization of cluster crystals with an intriguing structural diversity.

In the phase diagram of G1-P-G1 (Fig. 6a), a stable phase (blue region), which retains the diffraction features of a BCC structure and shows unexpected optical anisotropy (BCC$_{LC}$), is observed at DNA densities above the ones where the cluster BCC crystal phase (red region) occurs. This birefringent phase is stable over a remarkable wide range of concentrations and temperatures. In addition, the phase transition from BCC-to-BCC$_{LC}$ can also be reversibly driven through the alteration of temperature (at DNA concentration $c = 300.8$ mg/ml in Fig. 6a, see also Supplementary Fig. 4), in accordance with the thermodynamic stability of this phase. The concentration-dependent series of SAXS profiles acquired at 38 °C (Fig. 7a), together with the corresponding depolarized images of the samples (top panel inset of Fig. 7b) offer a qualitative insight into the possible molecular origin of the BCC$_{LC}$ and its clustering character (a more detailed discussion is presented in Supplementary Note 4). A molecular packing scenario for the BCC$_{LC}$ phase based on orientationally ordered ellipsoidal-like clusters occupying the sites of the BCC lattice is schematically shown in Fig. 7c.

Taken together, our work on self-assembly of purely repulsive all-DNA dendritic-based triblock unambiguously demonstrate the experimental realization of the long-anticipated equilibrium cluster crystal structure at high densities. The experimental discovery of this unconventional crystalline state of matter is based on five pillars which constitute the key and unique identifying properties of cluster crystals, namely: the absence of micellar aggregates in dilute conditions; the emergence of clusters in the fluid phase at $c \lesssim c^*$; the stability of the clusters in the absence of attractions; the crystallization of the cluster fluid into a cluster crystal; and finally the crystal's remarkable adaptability in crowded conditions, as reflected by the density-independent lattice constants (top panel of Fig. 7b) and the scaling $N_{occ} \propto c$ (bottom panel of Fig. 7b), that holds over a broad concentration range. The latter is the most prominent hallmarks of cluster crystals predicted by theory[1–4] and hereby experimentally confirmed.

An intriguing aspect of this study is that the building block design, reliant on the block copolymer paradigm, confers our DNA-based triblock with temperature-responsiveness, resulting to cluster crystal structures with unexpected optical anisotropy and self-assembly transition pathways that can readily be controlled by altering the temperature. This confluence of self-assembly approaches from block copolymers[62] and DNA nanotechnology[63] has already been shown to be capable of furnishing temperature-regulated nanoscale structures with high levels of self-assembled structural complexity[46]. Given its versatile and robust character, the synthetic scheme reported here can easily be applied to any kind of all-DNA nanoscale architecture. We foresee that the present developments in the design and

construction of DNA nanostructures with arbitrary complexity in the research field of structural DNA nanotechnology can provide unprecedented freedom in cluster crystal engineering through careful design of the tethered all-DNA geometry, and therefore opens up the possibility of steering the macroscopic properties of the system in a predictive manner.

## Methods

**Synthesis of DNA-polymer hybrids.** Custom, phosphorylated and dibenzylcyclooctyne (DBCO)-modified oligonucleotides were purchased from Biomers and purified by HPLC. The DNA sequences used for the fabrication of all-DNA dendrons of first (G1) and second (G2) generation were adapted from previous work with a slight modification in order to suppress the base-stacking attraction between the dendrons' blunt-ends[43]. To assemble the reported DNA-based dendritic nanostructures, a strain-promoted alkyne-azide cycloaddition (SPAAC, copper-free click chemistry) reaction was employed[46]. To construct G1-P and G1-P-G1, a three-arm DNA junction (G1) having one arm terminated with a DBCO and a mono- or di-azide (N$_3$) end-functionalized Poly(2-oxazoline)-based copolymer Poxa were dissolved in 1xTris/Na buffer (10 mM Tris-HCl, pH 8.0, 150 mM NaCl) and homogenized, respectively. The optimal ratio of reacting compounds (DBCO/N$_3$) was found to be close to 4:1. The reactions are carried out at room temperature overnight. A detailed description of the azide-functionalized Poxa polymer synthesis procedure and its characterization is provided in Supplementary Methods 2–3.

For the fabrication of G2-P-G2, a core G1-P-G1 with its free-ends terminated with a phosphorylated non-palindromic 4-base single-stranded overhang (sticky end) is employed. This tetra-functional dendritic-polymer core was hybridized with four other G1 molecules with sticky-ends complementary to the G1-P-G1 in order to finally assemble the G2-P-G2. The post-assembly ligation of the G2-P-G2 is performed by the T4 DNA ligase (Promega). The three-arm DNA junctions involved in the construction of the G1-P, G1-P-G1 and G2-P-G2 were formed by hybridizing three partially complementary to each other synthetic ssDNA strands in a 1:1:1 stoichiometric ratio in 1xTE/Na buffer (10 mM Tris-HCl, pH 7.5, 0.1 mM EDTA and 150 mM NaCl)[42]. The final concentration was 15 µM for each strand. The DNA concentration was determined by measuring the absorbance at 260 nm with a micro-volume spectrometer (NanoDrop 2000). The oligo mixtures were cooled slowly from 90 °C to room temperature in 10 L water placed in a styrofoam box over 48 h to facilitate strand hybridization. The detailed construction scheme and sequence of DNA oligos used to assemble G1-P, G1-P-G1 and G2-P-G2 systems are listed in Supplementary Method 1. Nondenaturing polyacrylamide gel electrophoresis (PAGE) experiments were employed to confirm the successful assembly of G1, G1-P, G1-P-G1, and G2-P-G2. As shown in Fig. 1, the desired all-DNA and DNA-based dendritic constructs migrate as single sharp bands, with the dendritic-linear-dendritic triblock to demonstrate decreasing mobility with increasing generation.

**System parameters.** The radius of gyration ($R_g$) is obtained from small-angle X-ray scattering (SAXS) and hydrodynamic radius ($R_H$) is obtained from DLS. The corresponding values for the G1-P-G1 and G2-P-G2 are 6.4 nm and 9.8 nm for $R_g$; 4.9 nm and 7.9 nm for $R_H$, respectively (more details in Supplementary Note 1). The DNA overlap concentration ($c^*$) of the employed dendritic-linear-dendritic triblock (G1-P-G1 and G2-P-G2) was estimated using the equation $c^* = M_w/(\frac{4}{3}\pi R_H^3 N_A)$, where $M_w$ represents the DNA molecular weight of the triblock ($M_w^{G1-P-G1} = 58.09$ KDa, $M_w^{G2-P-G2} = 170.36$ KDa), $R_H$ the hydrodynamic radius at $T = 38$ °C, and $N_A$ Avogadro's number. The average cluster occupancy $N_{occ}$ is calculated using the relationship $N_{occ} = cN_A a_{BCC}^3/(M_w f)$, where $c$ is the total DNA concentration, $a_{BCC}$ the edge length of the BCC conventional unit cell, and $f$ is the number of clusters per unit cell ($f = 2$ for a BCC lattice). The molecular weight of the Poxa chains is 15.9 KDa for the G1-P-G1, G2-P-G2, and 19.6 KDa for the G1-P. The concentrated DNA solutions were prepared using 1xTris/Na buffer.

**Purification of G1-P-G1 and G2-P-G2.** To purify desired dendritic-linear-dendritic structures, reactions products were separated by size-exclusion chromatography (SEC) on Superdex 200 Increase 10/300 GL. The column was developed at the flow rate of 0.2 mL/min in 1xTris/Na buffer. The reaction mixture was diluted in the same buffer to the final concentration of 2.0−4.0 mg/mL and injected into the column by 0.3−0.4 mL portions per run. Two major peaks were collected and the corresponding fractions were analyzed by PAGE. The first peak corresponded to the DNA-based dendritic triblock and the second one to not react to three-arm DNA junctions. Fractions eluted from 10.0−11.0 mL were pooled together and concentrated by ultrafiltration (Amicon-Ultra 15 mL centrifugal filters). For the purification of G1-P, an anion exchange chromatography column HiPrep DEAE Fast Flow16/10 (GE Healthcare) was used[46].

**Light scattering (LS) experiments.** DLS and static light scattering (SLS) experiments were performed by employing an ALV goniometer setup equipped with a Helium-Neon laser operating at $\lambda = 632.8$ nm. The Brownian motion of the G1-P-

G1, G2-P-G2 and G1-P was recorded in terms of the time autocorrelation function of the polarized light scattering intensity, $G(q, t)$, employing an ALV-5000 multi-tau digital correlator. In DLS experiments, the intermediate scattering (field) function $C(q, t) = \sqrt{(G(q, t) - 1)/\beta}$ at several scattering wave vectors $q = (4\pi n/\lambda)\sin(\theta/2)$ is obtained, where $\beta$ is an instrumental factor related to the spatial coherence constant and depends only on the detection optics, $n$ the refractive index of the solvent, and $\theta$ the scattering angle. To analyze $C(q, t)$, an inverse Laplace transformation using the CONTIN algorithm was applied. The average relaxation time was determined from the peak of the distribution of relaxation times $L(\ln \tau)$[43] (see Supplementary Fig. 1a).

**Small-angle X-ray scattering (SAXS).** SAXS experiments were carried out at the high brilliance Galium Anode Low Angle X-ray Instrument (GALAXI) of the Jülich Center for Neutron Science (JCNS, Germany)[64]. A Dectris-Pilatus 1M detector with a resolution of $981 \times 1043$ pixels and a pixel size of $172 \times 172\ \mu m^2$ was employed to record the 2D SAXS scattering patterns. The 2D SAXS patterns were integrated using FIT2D software. Bragg peaks indexing in 1D SAXS profiles was performed using the Scatter computer program[65]. 1D SAXS profiles fitting was done using Jscatter[66] (more details on the fitting procedure are given in Supplementary Note 2). The DNA solution was thoroughly homogenized (up to 3 days for the more viscous samples) ensuring the absence of spatial concentrations gradients before loading into capillaries (2 mm thickness borosilicate, Hilgenberg) for SAXS experiments. The capillaries were sealed and stored at 4 °C for at least 1 month before being used for X-ray experiments. Long term stability and reproducibility were confirmed by repeating SAXS measurements on selected samples almost 6 months later (G1-P-G1 system, Supplementary Fig. 5).

## Data availability
The relevant data sets generated during and/or analyzed during the current study are available from the corresponding authors on reasonable request.

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

## Acknowledgements

We thank Prof. Valentin Gordeliy for providing the purification facilities. This work has been supported by the Deutsche Forschungsgemeinschaft (DFG) under grants STI 664/4-1; E.S. and JO 370/5-1; U.J. and by the Austrian Science Fund (FWF) under grant I 2866-N36; C.N.L.

## Author contributions

E.S. and C.N.L. conceived the project and designed research; E.S. synthesized the DNA-based dendritic nanostructures; N.J. and U.J. performed the polymer synthesis and characterization; T.B. performed the preparative SEC for the G1-P-G1 and G2-P-G2 purification; N.A. performed the Monte Carlo simulations; E.S. performed the experiments and purification of G1-P; E.K. and U.R. operated the SAXS beamline; R.B. performed the fittings in the 1D-SAXS profiles; J.K. G.D. contributed to initiating the project and data interpretation; E.S. and C.N.L. interpreted the data and wrote the manuscript. All co-authors commented on the manuscript.

## Competing interests

The authors declare no competing interests.
