## [Peer Review File · Nature Communications]

REVIEWER COMMENTS

Reviewer #1 (Remarks to the Author):

See attached.

Reviewer #2 (Remarks to the Author):

I have mixed feeling about this manuscript. On the one hand, I appreciate the relevance of the topic (soft-clusters), the excellent experimental work, the exquisite design of the particles, the clarity of the introduction. On the other, in some parts I find the reading quite disturbed by the breaks produced by the numerous references to supplementary material and by the absence of crucial basic information. Last but not least I have some doubts about the interpretation of the results.

Let me start by the last point. The authors design the "particle" by

decorating the two ends of a polymer (Poxa) with a DNA-made trimer (G1-P-G1)

and a "second generation" DNA construct (G2-P-G2). These particles, one can guess, experience some sort of "hydrophobic" attraction in the Poxa part and some sort of electrostatic repulsion steaming from the DNA part. These constructs are thus

quite reminiscent of an amphiphilic molecule. The authors discover that

these molecules do form first a cluster phase and then a cluster crystal (BCC and BCC_LC). I would have expected a much thorough discussion on the

conceptual differences between the soft-clustering discussed in the introduction and micellar clustering, to reassure the reader that the two concepts are different and that indeed, what they are observing is the soft-clustering one. Even the picture that they make of the cluster does resemble the typical representation of a micelle.

I agree with the authors that the q -independence of the peak position with concentration is a smoking gun for the soft-clusters, but I am not convinced that it is unique to this clustering process.

I would expect that, for micelles composed by a small number of amphiphilic molecules, the optimal micelle size can fluctuate a lot, which would account for

a weak concentration dependence of the number of micelles in the crystal phase.

The only observation that I can see in favour of the soft-cluster is the absence of a critical micellar concentration, but again this could be due to the small micelle size.

Let me now be more explicit about what I mean with "absence of crucial basic information". Writing a manuscript like this, for a broad audience, one should not expect the readers to be aware of all the technical aspects. For example,

what is Poxa ? Not everybody can connect the information of a lower critical point with a relevant hydrophobic attraction. Also, a more physical definition of the units of concentration (mg of the construct or mg of Poxa or mg of DNA ?) and how mg/ml translates in physical units could make reading easier (the authors do provide a figure for the overlap concentration later on. It should be provided upfront). Salt concentration and salt effects are also very poorly described. Informing the reader about the screening length corresponding to the experimental conditions would help.

Concerning the supplementary figures and supplementary notes. Let me just give an example. Between lines 360 and 390 (30 lines) there are 12 "see Supplementary Fig" or "see Supplementary notes". I understand the importance of providing additional information, but at this level, reading (and understanding the basic message) become almost impossible.

Let me now go to the positive aspects of this manuscript:

First, the topic is quite important. Soft clusters, discovered in soft-matter, have shown to be a quite important topic in fields as different as matter-light interaction, supersolids, and possibly biological applications. If the interpretation propose by the authors is correct, this would be the first experimental evidence of soft-clusters and soft-crystals in soft matter. A finding that , in my opinion, would by itself deserve

to be published in the parent "Nature" journal. The present manuscript is also very relevant in demonstrating how to exploit theory-informed design to conjugate DNA-made particles with hydrophobic molecules, expanding the already rich arsenal of DNA-bricks (a reference to the Chemical Review article by Dong et al and to the work of DNA-made particles conjugated with cholesterol by De Michele et al would be useful in this respect).

The experimental work is also excellent. It is not easy from a technical point of view

to perform scattering measurements on these samples and the corresponding literature, even for simple DNA-made particle is still scarce. The authors are able to follow in concentration and in temperature the self-assembly process and detect unambiguously the structural changes.

In summary, as I said, I have mixed feeling about this manuscript. At the same time I am really open minded. If the revised version properly answer all my doubts

and a detailed discussion of the difference between micelle and soft-cluster is provided, I will be more than happy to support publication.

Minor points:

113. Is 150 mM the salt concentration used in all experiments ? What is the corresponding screening length ?

165. Which are the "intriguing phase transition pathways" ?

174. Absence of micellization... "demonstrated" is too much !

202. Aqueous saline solution. Please be precise.

223. I am not 100 per cent convinced about the statement in this line. A more tentative phrasing could be reasonable.

265. Please be more precise about what is consistent with theory

--- The G1-P does not show any clustering. Still, if the hydrophobic attraction is strong (high T) one would also expect this construct to form aggregates. Please discuss.

--- What happens if one screens almost completely the electrostatic repulsion ?

REPLY TO REVIEWER 1

Comments for Author

The authors describe the first experimental realization of a soft matter cluster crystal. Based on a vast collection of theoretical investigations into the formation of cluster crystals, the authors utilize DNA-based triblock copolymers comprised of a poly(2-oxazoline) core with non-sticky dendritic DNA on either end to meet the theoretical requirements. They demonstrate a clear concentration and temperature dependence that results in first, cluster formation, and second, cluster crystal formation, as concentration or temperature are increased. The authors further outline the phase space that results from systematically changing these parameters and demonstrate how the size of the tri-block copolymer impacts the lattice constant.

The subject matter (cluster crystals) is of interest to the self-assembly and soft condensed matter communities. The experimental realization of a theoretically anticipated assembled state, as well as temperature-regulated phase behavior and some of the unexpected optical properties stemming from underlying anisotropy in clustering, should draw the interest of researchers from other fields. The design and use of an overall repulsive building block (see more below) for accessing novel condensed phases could spur new thinking in soft matter assembly. However, I have significant concerns about some of the claims and experimental evidence (see details below). In addition, the clarity and quality of presentation of this manuscript should be improved; there are significant portions of the work that are difficult to understand or are confusing. Such issues should be addressed prior to this work being reconsidered for publication in *Nature Communications*.

Reply: We thank the reviewer for the positive assessment of our work and for stating clearly that we are describing “the first experimental realization of a soft matter cluster crystal.” The reviewer further finds that “[t]he experimental realization of [the cluster crystals] as well as temperature-regulated phase behavior and [their] unexpected optical properties [...] should draw the interest of researchers from other fields”. We very much appreciate their statement that “the design of a new building block for accessing novel condensed phases could spur new thinking in soft matter assembly.” At the same time, the reviewer raises concerns regarding the clarity and quality of the presentation. We are thankful for the constructive criticism that follows and we are glad to reply to the comments raised by the reviewer on a point-by-point basis below. Addressing them has also led to thorough revisions of the manuscript, as explained in what follows.

Questions about the experimental results:

The authors assert that the DNA dendron-Poxa block copolymers possess a net repulsive interaction; however, experimental or theoretical evidence is not presented to support this claim. As mentioned by the authors, the Poxa block provides an attractive interaction potential, which could override the electrostatic repulsion of DNA under certain temperatures or salt conditions. This is especially important as the net-repulsive interaction is one of the key design requirements for realizing cluster crystals.

Reply: We thank the reviewer for bringing up this point that helps us clarify an important issue, which also relates closely to a remark raised by reviewer 2 regarding the difference between micellization and cluster formation. The reviewer points out that Poxa, at sufficiently high temperatures, becomes non-soluble to water and therefore the effective attractions between the Poxa monomers (hydrophobicity) may result into net attractions between the building blocks. There is ample theoretical evidence and direct experimental proof that this is not the case and that the clustering phenomena observed are not driven by attractions. We present our arguments below.

- To begin with, let us briefly clarify that what we called “a net repulsive interaction” refers to the effective interaction (free energy) $\Phi_{\text{eff}}(r)$ between our G1-P-G1 and G2-P-G2 complexes, where r stands for the distance between their centers of mass. This quantity can be calculated, e.g., following methods similar to those applied

in [Mladek *et al.*, Phys. Rev. Lett. **100**, 028301 (2008)], Reference [23] of the original manuscript. By its nature, $\Phi_{\text{eff}}(r)$ also contains very strong entropic contributions. Moreover, it is useful to clarify the meaning of the notion “repulsive” in this context. In the strict sense, a repulsive $\Phi_{\text{eff}}(r)$ would mean an effective potential for which $\Phi'_{\text{eff}}(r) \leq 0$ for all r , resulting into a repulsive force. In the broader sense, it means $\Phi_{\text{eff}}(r) \geq 0$ for all r , meaning that bringing the centers of mass of the particles to any finite separation R entails a free energy cost in comparison to infinite separations. These two cases are exemplarily shown in Fig. 1 below, panels (a) and (c), respectively, which is reproduced from the paper by Mladek *et al.* above.

FIG. 4 (color online). The effective potentials $\Phi_{\text{eff}}(r)$ of typical amphiphilic dendrimers [(a),(c)] and their FTs $\tilde{\Phi}_{\text{eff}}(q)$ [(b),(d)], showing negative parts. The blue dashed line denotes the theoretical result (see text), and the green lines are fits to the simulation data. (a) and (b) pertain to dendrimer D_1 , (c) and (d) to D_2 . The insets in (a) and (c) feature simulation snapshots of the respective dendrimers.

FIG. 1. The figure and its caption are reproduced from [Mladek *et al.*, Phys. Rev. Lett. **100**, 028301 (2008)]. Two different types of amphiphilic dendrimers result into effective interactions that are repulsive in the strict [(a)] and the broad [(c)] senses, respectively. In both cases, the Fourier transform $\tilde{\Phi}_{\text{eff}}(k)$ contains negative parts, thus the potential is of the Q^\pm -type, leading to cluster formation.

- Our DNA-Poxa-DNA constructs (building blocks) are indeed inspired by the design ideas put forward in the manuscript above. There, core (C) and shell (S) monomers are of different types: the C-C microscopic interactions are attractive but both C-S and S-S are repulsive, resulting into suitable Q^\pm potentials. For the constructs at hand, the C-monomers are the Poxa segments whereas the S-ones are DNA base pairs; the latter being electrically charged, they interact by an electrostatic, repulsive potential, screened by the salt present in the solution. The C-S interactions are also repulsive, as the Poxa and the DNA segregate due to their chemical dissimilarity, thus one has to focus on the C-C interactions in more detail.
- We know that bare Poxa features a $T_{\text{LCST}} \cong 33^\circ\text{C}$; at lower temperatures, Poxa is water-soluble and thus the C-C microscopic interactions there (the solvent having been integrated out) are repulsive. Accordingly, for such temperatures the effective potential $\Phi_{\text{eff}}(r)$ is indeed repulsive in the strict sense. As can be seen from Figs. 4(a),(b) of our manuscript (original submission) as well as from the scattering data in Fig. 3(b)(G1-P-G1) and Supplementary Fig. 3(b) (G2-P-G2), both the cluster fluid and the cluster crystal are already stable at temperatures much lower than T_{LCST} , therefore their stability is not due to putative attractions between our building blocks but rather due to the Q^\pm character of the potential.
- For temperatures exceeding T_{LCST} , effective attractions between Poxa-units arise and thus they could, in principle, induce also attractions in $\Phi_{\text{eff}}(r)$. However, Poxa’s amphiphilicity in Poxa-based conjugates is

strongly dependent on the architecture of the block copolymer and the physico-chemical properties of the blocks that composed it. This is manifested in a striking way in our SAXS data, where in the temperature range $33^\circ\text{C} \leq T \leq 38^\circ\text{C}$ we consider, the cluster crystals keep maintaining the same structural characteristics as the ones obtained for $T \leq 33^\circ\text{C}$ (see in the original submission, the temperature-dependence of Bragg reflections (red vertical solid lines) in Fig. 3(b) and Supplementary Fig. 3(b)); in which the $\Phi_{\text{eff}}(r)$ is purely repulsive, as argue above. If any attractions appear at all, then they must be only in the broad sense mentioned above. A relevant note is added in the revised version (lines 443-455) in order to highlight the absence of a micellization mechanism in the reported DNA-based dendritic building blocks.

- Any attractive contribution in the sense of a strongly negative part of the effective potential would have much more drastic effects in the self-organization scenarios of the system at hand. In particular, and making now also a connection to the remarks expressed by reviewer 2, this would lead to the formation of big aggregates (micelles) at very small concentrations, and indeed much lower than the overlap concentration c^* . To demonstrate that this is an entirely different class of phenomena, we have performed a number of additional experiments with an array of diverse types of newly synthesized DNA-Poxa building blocks, establishing that in the presence of attractions micellar aggregates form at temperatures slightly exceeding T_{LCST} . The complete absence of such occurrences for the building blocks employed in this work (G1-P-G1 and G2-P-G2) offers additional corroboration to our claims regarding the repulsive character of their effective interactions.
- To avoid repetitions, we present those additional results in our reply to reviewer 2. We are thankful to both reviewers for emphasizing the importance of this point, and we have now added an extended, separate section in our revised version entitled “Absence of a micellization mechanism in DNA dendritic-based triblocks” (lines 182-280); in which the evidence is presented in the main paper (some parts of it were before at the Supplementary Information) and a detailed discussion regarding the nature of the effective potential and the absence of micellization is put forward.

Why is there an extra base in the Y1c strand?

Y1c : 5'-(T5) AGG CTG ATT CGG TTC ATG CGG ATC CA-3'

Y1a : 5'-DBC0-TEG-(T5) TGG TCC GCA TGA CAT TCG CCG TAA G-3'

Reply: We thank the reviewer for bringing this to our attention. The Y_{1c} strand sequence is correct. However, there was a typographic error in the Y_{1a} and Y_{2a}^P strands sequence. The 4th base (A) from the 5'-end is missing (counting from the end of the PolyT and sticky-end sequence, respectively). In the revised version of the “Supplementary Information” the missing base is highlighted in blue.

Though the DNA was purchased from a supplier, the mass values (calculated and experimental) of the DNA strand sequences are missing.

Reply: The theoretical molecular mass (M_w^{theor}) of each individual DNA strand and the corresponding experimental value (M_w^{exper}) as determined by Maldi-TOF mass spectrometry are listed below. These values were provided by the supplier (Biomers) and they are included in “Supplementary Method 1” on the revised version of our manuscript.

$$\begin{aligned} \text{Y}_{1a}: M_w^{\text{theor}} &= 10063 \text{ g/mol}, M_w^{\text{exper}} = 10081 \text{ g/mol} \\ \text{Y}_{1b}: M_w^{\text{theor}} &= 9461 \text{ g/mol}, M_w^{\text{exper}} = 9465 \text{ g/mol} \\ \text{Y}_{1c}: M_w^{\text{theor}} &= 9523 \text{ g/mol}, M_w^{\text{exper}} = 9529 \text{ g/mol} \\ \text{Y}_{1b}^{\text{P}}: M_w^{\text{theor}} &= 9256 \text{ g/mol}, M_w^{\text{exper}} = 9264 \text{ g/mol} \\ \text{Y}_{1c}^{\text{P}}: M_w^{\text{theor}} &= 9318 \text{ g/mol}, M_w^{\text{exper}} = 9322 \text{ g/mol} \\ \text{Y}_{2a}^{\text{P}}: M_w^{\text{theor}} &= 9287 \text{ g/mol}, M_w^{\text{exper}} = 9293 \text{ g/mol} \end{aligned}$$

A previous simulation showed that an FCC phase is preferred over a BCC phase for amphiphilic dendrimers (J. Chem. Phys. **144**, 204901 (2016)). Why is an FCC phase not observed in this system?

Reply: We thank the reviewer for this remark. The results presented in the reference mentioned in the comment have been obtained for a particular model of amphiphilic dendrimers, in which some specific forms of Morse-type interactions for the core- and shell-monomers have been arbitrarily introduced; it does not describe the system at hand. Accordingly, there is no general argument establishing that one crystal type to be more stable than another, although in all cases in which cluster crystals have been theoretically studied by means of generic models, the two

competing ones have been the FCC and the BCC. Interestingly enough, a generic density-functional approach predicts the BCC crystal to be the most stable structure [Likos *et al.*, J. Chem. Phys. **126**, 224502 (2007)], Reference [7] of the original manuscript, under certain simplifying assumptions. We emphasize, however, that the most stable structure is determined by details of the systems and their interactions.

The cluster crystal assembly conditions are unclear - was the solution salted, buffered, or neither?

Reply: This information is provided in the “Methods” section (“System parameters” subsection (line 591 in the original submission). The concentrated DNA solutions were prepared using 1xTris/Na buffer (10 mM Tris, pH 8.0, 150 mM NaCl). We agree with the reviewer that this information is important, and, therefore we add a relevant note in the end of “Building blocks design” section (revised version, lines 169-171) for easy and faster access.

In Figure 3a, the BCC_{LC} cluster 1D SAXS profile seemingly shows a low q (between 0.1 and 0.2) feature. Is that expected?

Reply: This very broad and weak in intensity feature in our 1D-SAXS profile is related to unmasked “hot” isolated pixels due to extra scattered intensity around the beam stop. In addition, the temperature-dependent SAXS profiles of the same system (G1-P-G1, $c = 300.8$ mg/ml) in the Supplementary Fig. 5 (original submission) do not exhibit this low- q signal. Therefore, the above-mentioned SAXS feature in Fig. 3a can not be associated with the BCC_{LC} cluster phase.

The authors described the phase at 337.7 mg/mL formed by G1-P-G1 as a “mechanically unstable phase” What does that mean, and what are the implications of this?

Reply: This concentration is located in the structurally unidentified orange region of the phase diagram of G1-P-G1 (Fig. 4a) with the latter surrounded by the BCC_{LC} cluster, HPC and G_{LC} . In contrast to the SAXS signature of the liquid crystalline glass-like phase (G_{LC} , see top profile in Fig. 4c), the 1D-SAXS profile corresponding to the unidentified orange region (second profile from the top in Fig. 3a, original submission) pertains almost all the higher order reflections from the BCC crystal. However, their relative positions to primary and most intense reflection are distorted. We speculate, that this unidentified phase is related to pre-transitional effects across the concentration-dependent BCC_{LC} -to-HPC and BCC_{LC} -to- G_{LC} transitions, and, therefore we have characterized it as a “mechanically unstable phase”.

Comments on presentation / clarity:

This manuscript would benefit from significant restructuring and rewriting. For example, the significance of cluster crystals is not clear upon a first reading. The figures were also not optimally arranged. The authors should consider showing a scheme of the dendritic block copolymer organizing into clusters and cluster crystals in the early portion of the manuscript. In addition, many sentence structures and discussions do not meet the level of rigor needed for publication in *Nature Communications*. Included below are two examples from the abstract:

Reply: We thank the reviewer for these suggestions and we have indeed made changes in the presentation and the organization of the manuscript to emphasize the importance of cluster crystals. We would like to point out that the manuscript was originally submitted to *Nature* (screened by the editor and transferred to *Nature Communications*), which has a more restrictive set of rules regarding length, therefore on the one hand we shortened the text and on the other we had to place some relevant material in the Supplementary information, which probably has interrupted the flow of the text. We hope that the revised version has improved in this sense.

1) “Cluster crystals are spontaneously forming periodic structures with lattice sites occupied by several, overlapping building blocks, and their site occupancy is a fluctuating quantity whose expectation value depends on thermodynamic conditions.”

The phrase “spontaneously forming” serves little purpose in the context of this sentence. The description of a multi-occupancy also is unnecessary.

Reply: We used the phrase “spontaneously forming” to emphasize self-assembly without additional manipulations but we have removed it from the revised version. On the other hand, we find the description of multi-occupancy to be necessary since not all readers are familiar with the concept of cluster crystals (or their quantum analog of supersolids) and we believe that this is helpful to them.

2) “Here we show that suitably tailored DNA-based dendritic nanostructures give rise to thermodynamically stable, well-controlled cluster crystals for a broad range of temperatures and concentrations at ambient conditions. Exploiting the base-pair selectivity of the DNA and the available toolkit from synthetic polymer chemistry, we fabricate dendritic-linear-dendritic tri-blocks composed of a thermosensitive water-soluble polymer and nanometre-scale all-DNA dendrons of the first and second generation, to engineer effective interactions necessary for the stabilization of this unusual crystalline state of matter”

Redundant adjectives, especially without relevant contexts: E.g., suitably tailored, well-controlled.

Reply: The adjectives used in the abstract are not at all unusual in such a context, since the abstract provides a summary of the content of the manuscript, the precise description of the work done being left for the main body of the paper. The adjectives are not redundant, in our view, the DNA-based constructs are indeed suitably tailored for the purpose of forming cluster crystals which are indeed well-controlled via concentration- and temperature changes. However, due to specific editorial requests for resubmission (abstract shortening), a significant part of the above-mentioned text is not included in the revised version.

Contradictory word choices: E.g. a broad range of temperatures at ambient conditions. Ambient conditions imply temperatures reasonably close to room temperature.

Reply: Again, this is an issue of semantics. In soft matter science, the breadth of temperatures one can reach is usually much more limited than in other realms of natural science and it ranges around room temperature. Nevertheless, for a system that employs water as solvent, covering a temperature range from $5\text{ }^{\circ}\text{C} \leq T \leq 40\text{ }^{\circ}\text{C}$ can still be called “a broad range of temperatures”.

Distracting phrases: E.g. “Exploiting the base-pair selectivity of the DNA and the available toolkit from synthetic polymer chemistry.” In an abstract with limited words, one can simply state the structure synthesized, especially when new synthetic strategies were not employed.

Reply: We agree with the reviewer, that “new synthetic strategies” in a broad sense are not employed. The suggested phrase is removed from the revised version. However, it should be stressed that the construction of the reported DNA-based dendritic nanostructures is novel and posed new challenges which have not been addressed before, ranging from the specific positioning of thermoresponsive polymer into the all-DNA dendritic scaffold, to the sample purification, and scaling up of the DNA-polymer hybrids mass.

REPLY TO REVIEWER 2

Remarks to the Author

I have mixed feeling about this manuscript. On the one hand, I appreciate the relevance of the topic (soft-clusters), the excellent experimental work, the exquisite design of the particles, the clarity of the introduction. On the other, in some parts I find the reading quite disturbed by the breaks produced by the numerous references to supplementary material and by the absence of crucial basic information. Last but not least I have some doubts about the interpretation of the results.

Reply: We thank the reviewer for their detailed report and the constructive criticism that follows, which has helped us make our message more clear and precise. We appreciate very much the reviewer’s remarks about “[...] the relevance of the topic (soft-clusters), the excellent experimental work, the exquisite design of the particles, [and] the clarity of the introduction” of our manuscript. At the same time, the reviewer raises concerns about the breaks produced by the references to the supplementary material (caused by constraints in length at original submission, please see below), by absence of information (now provided in a more direct and clear manner in the revised version) and about the interpretation of the results (answered in detail in what follows).

Let me start by the last point. The authors design the ”particle” by decorating the two ends of a polymer (Poxa) with a DNA-made trimer (G1-P-G1) and a ”second generation” DNA construct (G2-P-G2). These particles, one can guess, experience some sort of ”hydrophobic” attraction in the Poxa part and some sort of electrostatic repulsion steaming from the DNA part. These constructs are thus quite reminiscent of an amphiphilic molecule. The authors discover that these molecules do form first a cluster phase and then a cluster crystal (BCC and BCC_{LC}). I would have expected a much thorough discussion on the conceptual differences between the soft-clustering discussed in the introduction and micellar clustering, to reassure the reader that the two concepts are different and that indeed, what they are observing is the soft-clustering one. Even the picture that they make of the cluster does resemble the typical representation of a micelle. I agree with the authors that the q -independence of the peak position with concentration is a smoking gun for the soft-clusters, but I am not convinced that it is unique to this clustering process. I would expect that, for micelles composed by a small number of amphiphilic molecules, the optimal micelle size can fluctuate a lot, which would account for a weak concentration dependence of the number of micelles in the crystal phase. The only observation that I can see in favour of the soft-cluster is the absence of a critical micellar concentration, but again this could be due to the small micelle size.

Reply: The reviewer raises here an important point, namely that of the distinction between the well-known micellization e.g., block copolymers in selective solvents, polymeric amphiphiles and small molecule amphiphile surfactant systems) and the cluster formation we report here. We have already dedicated a long paragraph on this topic in the original submission (lines 167-185) but since part of the evidence for the difference between the two was in the Supplementary Information (original submission, Supplementary Figure 1 and Supplementary Note 1), it is possible that the clarity of the arguments has been compromised. In any event, the remarks of the reviewer (as well as the related question of reviewer 1 as to whether the overall effective interaction between the building blocks is repulsive) has motivated us to perform additional work in two directions, namely:

- Perform additional synthesis and experiments with two different types of buiding blocks comprising a rodlike double-stranded DNA (dsDNA) terminated by a PolyT dangig tail and a Poxa segment that are covalently linked, namely: [dsDNA-Poxa-dsDNA] triblocks, coded R-P-R in what follows, as well as [dsDNA-Poxa] diblocks, coded R-P in what follows. The R-block contains a 26-base-pair duplex which in length corresponds approximately to the width of G1. Together with the G1-P-G1 and G1-P blocks, these constitute an array of systems with different ‘protective hydrophilic blocks’ (dendritic G1 vs. rodlike DNA) and/or different degree of protection (on both sides, G1-P-G1, R-P-R, vs. only on one side, G1-P, R-P, of the Poxa polymer). We will discuss below the systematic differences in the ensuing aggregation behavior of the same, which clearly demonstrates the absence of micellization for the G1-P-G1 system at hand.
- We have thoroughly rewritten the section on micellization vs. cluster formation (“Absence of a micellization mechanism in DNA dendritic-based triblocks” section in the revised manuscript, lines 182-280), including now

new data (temperature-dependent hydrodynamic radius of G1-P-G1 and G1-P in dilute solutions, Fig. 2 in the revised version) as well as information that was previously in the Supplementary information.

We employ the terms *micellization/micelle formation*, consistently with broadly employed usage, to denote the formation of aggregates between entities that contain at least two blocks with different solubility properties, *at concentrations vastly smaller than the overlap value c^** . We provide in Fig. 2 below two characteristic examples of such behavior which are most relevant to our DNA dendritic-based triblocks. Fig. 2a shows the critical micelle concentration (*cmc*, top panel) and the concentration-dependence of the size of aggregates (bottom panel) for an amphiphilic system in which the building blocks are Y-DNA junctions (such as our G1-dendron without the PolyT dangling tails) functionalized with the lipid molecule *N*-glutaryl phosphatidylethanolamine [Roh *et al.*, Small **7**, 74 (2011)]. The *cmc* resulting there (17 nM) is about 3 orders of magnitude smaller than the concentration in which we measured scattering intensities in dilute solutions (86 μ M) without any evidence of micellization for the G1-P-G1 system, see Fig. 3 below and the associated discussion.

FIG. 2. (a) Reproduced from Fig. 3 of [Roh *et al.*, Small **7**, 74 (2011)]: **Top panel**, spectroscopic determination of the critical micelle concentration for Y-DNA-lipid hybrids, resulting into liposome-like DNA-somes. The *cmc* is found to be 17 nM. **Bottom panel**, zeta potential and size distribution of DNA-some as a function of DNA-lipid concentration. (b) Reproduced from Fig. 7a and Fig. 9 of [Pan *et al.*, Langmuir **28**, 14347 (2012)]: **Top panel**, temperature-dependent SAXS profiles of a dilute solution of ssDNA-Pnipam. **Bottom panel**, temperature dependencies of the radius of gyration (R_g) and aggregation number (N_{agg}) as extracted from the analysis of the 1D-SAXS profiles. The SAXS data from dilute ssDNA-Pnipam solutions show the formation of (spherical) aggregates at LCST ($T_{LCST} \cong 35^\circ\text{C}$). The concentration of DNA-polymer hybrids was 2.0 mg/ml (43.7 μ M).

Fig. 2b shows the thermoresponsive micellization of an amphiphilic linear diblock copolymers consisting of single-stranded DNA (ssDNA) and poly(*N*-isopropylacrylamide) (Pnipam). Pnipam is a thermoresponsive polymer with similar LCST to our Poxa. In this work [Pan *et al.*, Langmuir **28**, 14347 (2012)], the thermo-triggered micellization of ssDNA-Pnipam was examined by DLS and SAXS. The top panel of Fig. 2b presents the temperature-dependent SAXS profiles for aqueous solutions of ssDNA-Pnipam at concentration of 2.0 mg/ml (43.7 μ M). As the temperature increased up to around Pnipam's LCST ($T_{LCST} \cong 35^\circ\text{C}$), the scattering intensity at lower scattering angles increased abruptly and two distinct oscillations appear in the 1D-SAXS profiles. Both are indicative of the formation of micellar aggregates consisting of hydrophobic Pnipam core surrounded by hydrophilic ssDNA. The corresponding temperature-dependence of the radius of gyration (R_g) and aggregation number (N_{agg}) are depicted in the bottom panel of Fig. 2b. For this system, the *cmc* is reported to be around 0.015 mg/ml (0.33 μ M). The SAXS and DLS measurements (original

submission, Supplementary Fig. 1a,b,d) on dilute solutions of G1-P-G1 ($c = 2.5 - 5.0 \text{ mg/ml} = 43.0 - 86.1 \mu\text{M}$) clearly demonstrate the absence of such temperature-dependent self-assembly behavior, even at temperatures well-above the T_{LCST} of Poxa.

Broadly speaking, this micellization behavior is typical of Janus-type molecules for which one block is solvophilic and the other solvophobic. As mentioned above, we have extended our study of micellization behavior to include now the R-P-R and the R-P molecules and we summarize the results in Fig. 3 below. It becomes evident that diblocks, such as the G1-P and the R-P readily form micellar aggregates at temperatures above $T_{\text{LCST}} \cong 33^\circ\text{C}$ of the Poxa, whereas the two triblock molecules, P-R-P and G1-P-G1 are much more stable against coagulation even deep into the temperature regime in which bare Poxa is hydrophobic. The additional stabilizing effect of the dendritic structure (G1 vs. R) can also be seen, as the former shows practically no sign of aggregation all the way up to 55°C , where a doubling of the size of the P-R-P aggregates can be observed.

FIG. 3. (a) Schematics of the triblock and diblock amphiphilic copolymers employed for study of the micellization behavior of DNA-polymer hybrids with different macromolecular architecture. From top to the bottom: G1-P-G1, G1-P, R-P-R (dsDNA-Poxa-dsDNA) and R-P (dsDNA-Poxa). (b) Temperature-dependent static light scattering (SLS) intensity (left-axis, circles) at a scattering angle of $\theta = 90^\circ$, and hydrodynamic radius (R_H , right-axis, triangles), normalized with respect to their values at $T = 15^\circ\text{C}$, of dilute G1-P-G1 (black curves), G1-P (red curves), R-P-R (green curves) and R-P (brown curves) aqueous solutions (with DNA concentration c equal to 5.0 mg/ml ($86.1 \mu\text{M}$), 3.0 mg/ml ($103.3 \mu\text{M}$), 4.0 mg/ml ($102.3 \mu\text{M}$) and 2.5 mg/ml ($127.9 \mu\text{M}$), respectively) using 1xTris/Na buffer (10 mM Tris, pH 8.0, 150 mM NaCl). The gray-zone indicates the temperature window where the concentrated G1-P-G1 and G2-P-G2 self-assembled phase behavior is investigated.

We further would like to emphasize that all experiments reported in our work were performed below 38°C gray area in Fig. 3b). In particular, at the broad range of temperatures covered that lies below the Poxa T_{LCST} , no attractions are present at all, and therefore (see also our reply to reviewer 1) the total effective potential $\Phi_{\text{eff}}(r)$ between our building blocks is repulsive, which implies that the cluster formation we observe there at $c \lesssim c^*$ is indeed the soft cluster formation predicted by theory and not a classical micellization process.

We are of course aware of the fact that in certain systems or models (e.g., [Capone *et al.*, J. Phys. Chem. B **113**, 3629 (2009)] or [Capone *et al.*, J. Phys.: Condens. Matter **23**, 194102 (2011)]), aggregates form at very high concentrations, several times the overlap value c^* . In those cases of *diblock copolymers*, however, there is no significant attraction between the blocks and the resulting solids lack the main characteristic of our soft cluster crystals, namely the concentration independent value of the lattice constant, which was correctly identified by the reviewer as the *smoking gun* of the cluster crystals predicted theoretically. Whether one calls such an aggregation also micelle formation can be, at the end of the day, just an issue of semantics. Physically, however, what characterizes cluster crystals is a combination of their unique, identifying properties, namely:

- The absence of cluster formation in dilute conditions;
- The emergence of clusters in the fluid phase at $c \lesssim c^*$;

- The stability of the clusters in the absence of attractions;
- The crystallization of the cluster fluid into a cluster crystal;
- The concentration-independence of the crystal lattice constant.

As our systems (both G1-P-G1 and G2-P-G2) fulfill all requirements and at the same time they are clearly distinct from related systems (G1-P or R-P) that follow very different aggregation pathways, we are confident that we have demonstrated the existence of soft cluster crystals for a broad range of concentrations and temperatures and for both building blocks G1-P-G1 and G2-P-G2.

We are thankful to the reviewer (as well as to reviewer 1 for their related remark) for insisting that these differences be discussed in more detail. We have created in the revised manuscript a separate section, entitled “Absence of a micellization mechanism in DNA dendritic-based triblocks” on this issue. In addition, we have brought the Supplementary Fig. 1b into the main text including the new data on the temperature-dependence of the hydrodynamic radius R_H (Fig. 2 in the revised version). Also, we modified Fig. 2 of the original submission by presenting, together with the simulations snapshots, the schematic of the proposed concentration-dependent mechanism for the G1-P-G1 clustering in the absence of attractions, and, finally, the G1-P-G1 clusters crystallization into a BCC cluster crystal structure (Fig. 3 in the revised version).

Let me now be more explicit about what I mean with “absence of crucial basic information”. Writing a manuscript like this, for a broad audience, one should not expect the readers to be aware of all the technical aspects. For example, what is Poxa? Not everybody can connect the information of a lower critical point with a relevant hydrophobic attraction. Also, a more physical definition of the units of concentration (mg of the construct or mg of Poxa or mg of DNA?) and how mg/ml translates in physical units could make reading easier (the authors do provide a figure for the overlap concentration later on. It should be provided upfront). Salt concentration and salt effects are also very poorly described. Informing the reader about the screening length corresponding to the experimental conditions would help.

Reply: We sincerely apologize for the frustration that some parts of the main text may have caused. In the revised version, a detailed information regarding the Poxa is provided in the start of the “Building blocks design” section (lines 109-117) together with the relevant citation to Supplementary Information. The concentration units refer to DNA concentration and was determined by spectrophotometry. In the original submission, with a few exceptions, the phrase “DNA concentration” was used extensively to indicate the concentration of the building blocks. The working buffer conditions were provided in the “Methods” section (“System parameters” subsection, line 591 in the original submission). In the revised version, we add a relevant note in the end of “Building blocks design” section (lines 169-171). The concentrated DNA solutions were prepared using 1xTris/Na buffer (10 mM Tris-HCl, pH 8.0, 150 mM NaCl). The corresponding Debye-Hückel screening length is 0.76 nm.

Concerning the supplementary figures and supplementary notes. Let me just give an example. Between lines 360 and 390 (30 lines) there are 12 “see Supplementary Fig” or “see Supplementary notes”. I understand the importance of providing additional information, but at this level, reading (and understanding the basic message) become almost impossible.

Reply: The reviewer is quite right. This fragmentation is the result of the fact that the manuscript has been originally submitted to *Nature*, which has severe restrictions on length, and it was then transferred without changes to *Nature Communications*. As a result, the whole discussion on the G2-P-G2-system had been delegated to the Supplementary Information, causing the reader the inconvenience of having to go back-and-forth. In the revised version we resubmit, all of the discussion on the G2-P-G2 (Supplementary Note 4 and Supplementary Fig. 3 in the original submission) is back in the main paper (lines 456-498 and Fig. 4c-d in the revised version), minimizing thus the excursions from the main paper to the Supplementary Information and back. In the revised version, a synopsis regarding the key and unique identifying properties of the discovered cluster crystals is added (“Conclusions” section, lines 603-617).

Let me now go to the positive aspects of this manuscript:

First, the topic is quite important. Soft clusters, discovered in soft-matter, have shown to be a quite important topic in fields as different as matter-light interaction, supersolids, and possibly biological applications. If the interpre-

tation propose by the authors is correct, this would be the first experimental evidence of soft-clusters and soft-crystals in soft matter. A finding that, in my opinion, would by itself deserve to be published in the parent "Nature" journal. The present manuscript is also very relevant in demonstrating how to exploit theory-informed design to conjugate DNA-made particles with hydrophobic molecules, expanding the already rich arsenal of DNA-bricks (a reference to the Chemical Review article by Dong et al and to the work of DNA-made particles conjugated with cholesterol by De Michele et al would be useful in this respect).

Reply: We thank the reviewer for their very positive statements and for acknowledging explicitly the degree of novelty as well as the importance of the work. We are convinced that we have indeed experimentally discovered soft matter supersolids and we are very much hoping to be able to persuade also the reviewers on this issue with the help of the additional evidence we present in the replies and in the revised manuscript. We are happy to cite relevant work mentioned by the reviewer (Refs. [56, 57] in the revised version).

The experimental work is also excellent. It is not easy from a technical point of view to perform scattering measurements on these samples and the corresponding literature, even for simple DNA-made particle is still scarce. The authors are able to follow in concentration and in temperature the self-assembly process and detect unambiguously the structural changes.

Reply: We thank the reviewer for acknowledging the technical points involved in the measurements performed.

In summary, as I said, I have mixed feeling about this manuscript. At the same time I am really open minded. If the revised version properly answer all my doubts and a detailed discussion of the difference between micelle and soft-cluster is provided, I will be more than happy to support publication.

Reply: We thank the reviewer for giving us the chance to strengthen our arguments and to submit a revised manuscript. We believe that clarity has been enhanced and hope that the arguments on the experimental discovery of the cluster crystals are now convincing.

Minor points

113. Is 150 mM the salt concentration used in all experiments? What is the corresponding screening length?

Reply: All the DNA solutions (concentrated and dilute) were prepared using 1xTris/Na buffer (10 mM Tris-HCl, pH 8.0, 150 mM NaCl). The corresponding Debye-Hückel screening length is 0.76 nm.

165. Which are the "intriguing phase transition pathways"?

Reply: The phase diagram of the G1-P-G1 in Fig. 4a (original submission) demonstrate a variety of different concentration- and temperature-induced phase transition pathways.

- c -dependent, $T = 38^\circ\text{C}$: BCC-to-BCC_{LC}-to-G_{LC}
- c -dependent, $T = 20^\circ\text{C}$: Cluster Fluid-to-BCC-to-BCC_{LC}-to-HPC
- T -dependent, $c = 255.7$ mg/ml: Cluster Fluid-to-BCC
- T -dependent, $c = 300.8$ mg/ml: BCC-to-BCC_{LC}
- T -dependent, $c = 356.2$ mg/ml: HPC-to-G_{LC}

In order to provide reviewer with easier access to the above-mentioned transitions, we indicate these with green rectangles in Fig. 4 below.

FIG. 4. Reproduced from Fig. 4a, original submission. The green rectangles indicate the different concentration- and temperature-induced phase transition pathways which are encountered in the phase diagram of G1-P-G1.

174. Absence of micellization... “demonstrated” is too much!

Reply: We hope that the new evidence and discussion we provide in the replies as well as in the revised manuscript will convince the reviewer that this statement is not an exaggeration but rather a statement of the state of affairs.

202. Aqueous saline solution. Please be precise.

Reply: The phrase “Aqueous saline solution” is replaced with “Aqueous solution containing 150 mM NaCl”.

223. I am not 100 per cent convinced about the statement in this line. A more tentative phrasing could be reasonable.

Reply: The origin of the q_{DNA} peak is addressed by earlier SAXS studies in DNA liquid crystals (LCs, Refs. 49-50 in original submission). It originates from a liquid-like positional order between neighboring, parallel DNA helices. In concentrated solutions of stiff DNA duplexes with sufficient degree of anisotropy, this Bragg reflection (q_{DNA} peak) is present through the isotropic-to-LC transition and becomes sharper and more intense as the concentration increases; with the wave vector q corresponding to the intensity maximum shifts to higher values. In the original submission, we used the words “DNA segments” instead of “DNA helices”. With the word “segments” we do not refer to correlations within the DNA helices, which they are expected to appear in $q > 10.0 \text{ nm}^{-1}$.

265. Please be more precise about what is consistent with theory

Reply: Consistent with theory is the fact that the latter predicts the gradual emergence of clusters in the fluid as the concentration increases, these clusters then increasing their population as density grows and ordering into a cluster crystal beyond a temperature-dependent solidification concentration. We have added this clarification at the appropriate position in the manuscript (revised version: lines 258-271 and schematic representation in Fig. 3).

— The G1-P does not show any clustering. Still, if the hydrophobic attraction is strong (high T) one would also expect this construct to form aggregates. Please discuss.

Reply: The reviewer is correct. In the G1-P system the clustering mechanism is absent. Indeed, the G1-P exhibits a thermo-triggered micellization at the LCST of Poxa as depicted in the temperature-dependence of its hydrodynamic radius R_{H} in Fig. 3 in our replies and in Fig. 2 in the revised manuscript. The DLS data presented in Supplementary Fig. 1a (revised version of Supplementary information) indicate the formation of large micellar aggregates with $R_{\text{H}} = 175.3 \text{ nm}$ and narrow distribution in size. The aggregates’s size suggests that G1-P diblocks self-assembled into

vesicle-type of micelles which may share intriguing structural similarities with the aggregates formed by Y-DNA-lipid amphiphilies reported in the work of Roh *et al.*, Small **7**, 74 (2011) (see schematic illustration and relevant sizes in Fig. 2a from our replies). However, further experiments are needed in order to obtain further insight regarding the packing of G1-P amphiphile within the aggregate structure. In response to the reviewer’s comment, and considering the points in our reply above, we have added a discussion on G1-P micellization in the “Absence of a micellization mechanism in DNA dendritic-based triblocks” section of the revised manuscript.

— What happens if one screens almost completely the electrostatic repulsion?

Reply: To demonstrate the influence of high salinity conditions on the cluster crystal formation, we have performed new temperature-dependent SAXS measurements on G2-P-G2 at DNA concentration $c = 220.0\text{ mg/ml}$ and salt concentration $c_{salt} = 500\text{ mM NaCl}$. The results are presented in the Fig. 5 below which contains for comparison the data from G2-P-G2 solution at similar DNA concentration with $c_{salt} = 150\text{ mM NaCl}$ (data taken from Fig. 4d, revised version). By increasing the temperature, both solutions, demonstrate a phase transition from (cluster) fluid to BCC crystal with $a_{BCC} = 28.2\text{ nm}$. The only difference is that increase of salt concentration shifts the transition to lower temperature. This is not surprising since the presence of salt in the Poxa solution will shift its LCST at lower temperature. However, the potential cluster crystals resilience against the addition of salt requires further investigation.

FIG. 5. Temperature-dependent SAXS profiles of the G2-P-G2 at DNA concentration $c \sim 220.0\text{ mg/ml}$ with different salt conditions. Blue profiles: $c_{salt} = 150\text{ mM NaCl}$; Black profiles: $c_{salt} = 500\text{ mM NaCl}$.

REVIEWERS' COMMENTS

Reviewer #1 (Remarks to the Author):

The authors have thoroughly addressed all of my concerns, and the clarity and presentation quality of this work have been significantly improved upon revision. In particular, the authors have, through additional experiments (Fig. 2), ruled out micelle formation as a potential mechanism for the observed assembly, strengthening their argument for the formation of cluster crystals in the absence of net attractive interactions between any building blocks. Given the quality and the impact of this work to the field of soft matter and beyond, I recommend publication of this manuscript in Nature Communications as is.

Reviewer #2 (Remarks to the Author):

I have read with great interest the revised version of the manuscript. I have appreciated the effort the authors have made to answer the questions consistently raised by the two reviewers. I have also appreciated the convincing new experimental results.

I believe that the new section "absence of a micellization mechanism in DNA dendritic-based triblocks" is very helpful to distinguish well-known micellization phenomena and the cluster formation. This section provides, if not a definitive answer, a significant support to the identification of the cluster phase as a different phenomenon from micelle formation.

The presentation of the material is now much smoother and the missing information is now provided. I am now more than happy to support publication of the present manuscript.

Replies to Referees - Manuscript ID NCOMMS-21-14920A**REPLY TO REVIEWER 1****Remarks to the Author**

The authors have thoroughly addressed all of my concerns, and the clarity and presentation quality of this work have been significantly improved upon revision. In particular, the authors have, through additional experiments (Fig. 2), ruled out micelle formation as a potential mechanism for the observed assembly, strengthening their argument for the formation of cluster crystals in the absence of net attractive interactions between any building blocks. Given the quality and the impact of this work to the field of soft matter and beyond, I recommend publication of this manuscript in Nature Communications as is..

Reply: We thank the reviewer for the positive assessment of our revision and for recommending publication of our work.

REPLY TO REVIEWER 2**Remarks to the Author**

I have read with great interest the revised version of the manuscript. I have appreciated the effort the authors have made to answer the questions consistently raised by the two reviewers. I have also appreciated the convincing new experimental results.

I believe that the new section "absence of a micellization mechanism in DNA dendritic-based triblocks" is very helpful to distinguish well-known micellization phenomena and the cluster formation. This section provides, if not a definitive answer, a significant support to the identification of the cluster phase as a different phenomenon from micelle formation.

The presentation of the material is now much smoother and the missing information is now provided. I am now more than happy to support publication of the present manuscript.

Reply: We thank the reviewer for the positive assessment of our revision and for recommending publication of our work.